# Weak and Strong Reversibility of Non-Deterministic Actions: Universality and Uniformity

**Primary Keywords:** *None*

## Abstract

Classical planning looks for a sequence of actions that transform the initial state of the environment into a goal state. Studying whether the effects of an action can be undone by a sequence of other actions, that is, *action reversibility*, is beneficial, for example, in determining whether an action is safe to apply. This paper deals with action reversibility of non-deterministic actions, i.e. actions whose application might result in different outcomes. Inspired by the established notions of weak and strong plans in non-deterministic (or FOND) planning, we define the notions of *weak and strong reversibility* for non-deterministic actions. We then focus on the universality and uniformity of action reversibility, that is, whether we can always undo all possible effects of the action by the same means (i.e. policy), or whether some of the effects can never be undone. We show how these classes of problems can be solved via classical or FOND planning and evaluate our approaches on FOND benchmark domains.

## Introduction

*Automated planning* is a subfield of Artificial Intelligence dealing with the problem of whether there exists a sequence of actions that lead from an initial state to some goal state of the environment (Ghallab, Nau, and Traverso 2004, 2016). In Fully Observable Non-Deterministic (FOND) planning, actions have non-deterministic effects, i.e. if an action is applied, one of the effects will (randomly) occur and hence the formalism can model the uncertainty associated with applying such actions (Cimatti et al. 2003).

An important research question concerns whether the effects of an action can be undone by means of other actions. In the literature, this problem is referred to as *action reversibility* (Morak et al. 2020). One of the main motivations of studying action reversibility is that actions identified as reversible, that is, those whose effects can be undone, are safe to apply. In other words, if a reversible action is applied, its application does not lead to a dead-end state (unless the action was applied in a state that already was a dead-end). This property is especially useful in online planning, where an agent plans and acts w.r.t. a short time horizon, as it might provide information about safe states (Cserna et al. 2018). Another possible benefit of action reversibility can be in post-planning plan optimization in which it might contribute to more efficient computations of redundant "action cycles" in plans (Med and Chrpa 2022). In FOND planning, the potential benefits of action reversibility also include determining whether recovery from undesirable effects is possible, which might lead to more efficient generation of strong cyclic plans (Muise, McIlraith, and Beck 2012; Camacho, Muise, and McIlraith 2016), or whether we can safely perform deterministic replanning in unknown states (cf. FF-replan (Yoon, Fern, and Givan 2007)).

A specific form of action reversibility involving the search for an *inverse action* has been investigated (Koehler and Hoffmann 2000; Chrpa, McCluskey, and Osborne 2012). The existence of a pair of inverse actions is a special case of action reversibility, where the reverse plan contains only one action. Eiter, Erdem, and Faber (2008) introduced the concept of *reverse plans* that corresponds to the notion of *uniform reversibility* established by Morak et al. (2020) referring to reversibility of action effects by the same means (i.e., by the same action sequence). A more general notion of (non-uniform) reversibility has been tackled by compilation of the problem into contingent planning (Daum et al. 2016) or logic programs (Faber, Morak, and Chrpa 2021). Recent work of Chrpa, Faber, and Morak (2021) investigates under which condition we can find universal reverse plans.

This paper studies the reversibility of non-deterministic actions in FOND planning, which is, to the best of our knowledge, the first work in this area. Inspired by the established notions of *weak and strong plans* (Cimatti et al. 2003) we propose notions of *weak and strong reversibility* that, in a nutshell, refer to "all effects might be undone" and "all the effects can always be undone", respectively. We then focus on a subclass of the problem concerning *universal uniform reversibility*, that is, whether we can always undo all effects of a (non-deterministic) action by the same (weak or strong) policy, or whether some of the effects can never be undone, i.e. they are *universally irreversible*. We show how these problems can be compiled into classical planning (for weak reversibility and irreversibility) or FOND planning (for strong reversibility). Our approaches are evaluated on a range of FOND benchmark domains.

## Preliminaries

We adopt terminology from the Simplified Action Structures (SAS⁺) formalism (Bäckström and Nebel 1995) and the FOND planning (Cimatti et al. 2003).

Let $v$ be a **variable** (or a **state variable**) and $dom(v)$ be its domain. A **fact $(v, x)$** is a pair that contains a variable $v$ and its value $x \in dom(v)$. The **set of all facts** over the set of variables $\mathcal{V}$ is denoted as $\mathcal{F}(\mathcal{V})$. A set $\Sigma \subseteq \mathcal{F}(\mathcal{V})$ is called a **variable assignment** over $\mathcal{V}$ if and only if for all $(v, x) \in \Sigma$ does not exist $y \in dom(v)$ such that $y \neq x$ and $(v, y) \in \Sigma$. The **value** of the variable $v$ in $\Sigma$, denoted as $\Sigma[v]$, is equal to $x$ if and only if $(v, x) \in \Sigma$. The **set of all variables** in $\Sigma$, denoted as $vars(\Sigma)$, is defined as $vars(\Sigma) = \{v \mid (v, x) \in \Sigma\}$. A variable assignment $\Sigma$ over the set of variables $\mathcal{V}$ is called a **complete variable assignment** if and only if $vars(\Sigma) = \mathcal{V}$. Otherwise, it is called a **partial variable assignment**. The **set of all complete variable assignments** over $\mathcal{V}$ is denoted as $\mathcal{S}(\mathcal{V})$.

An **action $a$** over a set of variables $\mathcal{V}$ is a pair $(pre(a), eff(a))$, where $pre(a)$ is a variable assignment over $\mathcal{V}$ representing the **precondition** of the action $a$, and $eff(a)$ is a non-empty set of variable assignments over $\mathcal{V}$ representing the **set of possible effects** of the action $a$. The action $a$ is called **deterministic** if and only if $|eff(a)| = 1$. A **determinisation** of an action $a$ with respect to the effect $e \in eff(a)$, denoted as $a_e^d$, is an action $a_e^d = (pre(a), \{e\})$. The set of variables related to an action $a$, denoted $vars(a)$, is the set of variables $vars(pre(a)) \cup \bigcup_{e \in eff(a)} vars(e)$. We say that an action $a$ is **applicable** in a variable assignment $\Sigma$ if and only if $pre(a) \subseteq \Sigma$. The **set of applicable actions** in the variable assignment $\Sigma$ from the set of actions $\mathcal{A}$ is denoted as $\alpha(\Sigma, \mathcal{A})$. The **application of a deterministic action** $a = (pre(a), \{e\})$ in a variable assignment $\Sigma$ such that $pre(a) \subseteq \Sigma$, denoted as $\gamma(\Sigma, a)$, is the variable assignment $\gamma(\Sigma, a) = \{(v, x) \in \Sigma \mid v \notin vars(e)\} \cup e$ (the application is undefined if $a$ is not applicable in $\Sigma$). Given a deterministic action $a = (pre(a), \{e\})$, we define $ha(a)$ as the variable assignment $\gamma(pre(a), a)$. It is easy to prove $ha(a)$ is the largest set of facts that satisfies $ha(a) \subseteq \gamma(\Sigma, a)$ for all variable assignment $\Sigma$ that $a$ is applicable in; for this reason, we say that $ha(a)$ **necessarily holds** after applying action $a$.

The **application of a (non-deterministic) action** $a$ in a variable assignment $\Sigma$, denoted as $\delta(\Sigma, a)$, is the set of variable assignments $\delta(\Sigma, a) = \{\gamma(\Sigma, a_e^d) \mid e \in eff(a)\}$.

A **FOND planning domain $\mathcal{D}$** is a pair $\langle \mathcal{V}, \mathcal{A} \rangle$, where $\mathcal{V}$ is a set of variables, and $\mathcal{A}$ is a set of actions over $\mathcal{V}$. An **SAS⁺ planning domain** is a pair $\langle \mathcal{V}^d, \mathcal{A}^d \rangle$, where $\mathcal{V}^d$ is a set of variables and $\mathcal{A}^d$ is a set of deterministic actions over $\mathcal{V}^d$.

A **state $s$** of the domain $\mathcal{D} = \langle \mathcal{V}, \mathcal{A} \rangle$ is a complete variable assignment over the set of variables $\mathcal{V}$.

A **FOND (resp. SAS⁺) planning task $\mathcal{T}$** is a triple $\langle \mathcal{D}, s_I, G \rangle$, where $\mathcal{D} = \langle \mathcal{V}, \mathcal{A} \rangle$ is a FOND (resp. SAS⁺) planning domain, $s_I \in \mathcal{S}(\mathcal{V})$ is an **initial state** and $G \subseteq \mathcal{F}(\mathcal{V})$ is a variable assignment representing a **goal**. When we refer to a planning task or a domain, we mean a non-deterministic task or domain, unless stated otherwise.

A sequence of deterministic actions $\pi = \langle a_1, \dots, a_n \rangle$, $\forall i \in \mathbb{N}, 1 \leq i \leq n : a_i \in \mathcal{A}^d$, is called a **plan** for the SAS⁺ planning domain $\mathcal{D}^d$. An **application of the plan** $\pi = \langle a_1, \dots, a_n \rangle$ in a variable assignment $\Sigma$, denoted as $\gamma(\Sigma, \pi)$, is a variable assignment $\gamma(\Sigma, \pi) = \gamma(\gamma(\Sigma, a_1), \langle a_2, \dots, a_n \rangle)$. If $\pi = \langle \rangle$, then $\gamma(\Sigma, \langle \rangle) = \Sigma$.

A state $s$ is **reachable** in SAS⁺ planning task $\mathcal{T}^d$ if and only if there exists a plan $\pi$ for the domain $\mathcal{D}^d$ such that $\gamma(s_I, \pi) = s$. Otherwise, $s$ is **unreachable**. We say that $\mathcal{T}^d$ is **solvable** if and only if some goal state $s_G \supseteq G$ is reachable in the planning task $\mathcal{T}^d$. Otherwise, $\mathcal{T}^d$ is **unsolvable**. A plan $\pi$ is called a **goal plan** of the SAS⁺ planning task $\mathcal{T}^d$ if and only if $\gamma(s_I, \pi) \supseteq G$.

A **policy $\Pi$** for the domain $\mathcal{D} = \langle \mathcal{V}, \mathcal{A} \rangle$ is a binary relation over the set of states $\mathcal{S}(\mathcal{V})$ and the set of applicable actions of $\mathcal{A}$, i.e., $\Pi \subseteq \{(s, a) \mid s \in \mathcal{S}(\mathcal{V}), a \in \alpha(s, \mathcal{A})\}$. The **set of all states related in $\Pi$** is the set $\sigma(\Pi) = \{s \mid (s, a) \in \Pi\}$. The **$n$-step application** of $\Pi$ in a state $s$, denoted as $\delta^n(s, \Pi)$, is the set $\delta^n(s, \Pi) = \bigcup_{s' \in \delta^{n-1}(s, \Pi)} \bigcup_{(s'', a') \in \Pi, s'' = s'} \delta(s', a')$ for $n \geq 1$ with $\delta^0(s, \Pi) = \{s\}$.

We say that $s'$ is **reachable** from $s$ with a policy $\Pi$ if and only if $s' \in \delta^i(s, \Pi)$ for some $i \geq 0$. We say that $s'$ is a **terminal state** for $\Pi$ with respect to a state $s$ if and only if $s'$ is reachable from $s$ with $\Pi$ and $s' \notin \sigma(\Pi)$. The **set of all terminal states** is denoted as $\tau(\Pi, s)$. Note that our policy definition as a binary relation (i.e., more actions might be associated with a single state) straightforwardly supports a combination of two (or more) policies together by a union.

A policy $\Pi$ is called a **weak goal policy** for $\mathcal{T}$ if and only if $\tau(\Pi, s_I) \cap \{s_G \in \mathcal{S}(\mathcal{V}) \mid G \subseteq s_G\} \neq \emptyset$. A policy $\Pi$ is called a **strong goal policy** for $\mathcal{T}$ if and only if $\tau(\Pi, s_I) \subseteq \{s_G \in \mathcal{S}(\mathcal{V}) \mid G \subseteq s_G\}$ and for each $s \in \sigma(\Pi)$ at least one state that satisfies the goal $G$ is reachable by $\Pi$. Note that our definitions coincide with Def. 2.10 of Cimatti et al. (2003). A task $\mathcal{T}$ is called **solvable** if and only if there exists a weak goal policy for task $\mathcal{T}$. Otherwise, it is called **unsolvable**.

Let $\Psi$ be a set of pairs $(\Sigma, a)$, where $\Sigma$ is a variable assignment, and $a \in \mathcal{A}$ is an action that is applicable in $\Sigma$. The set $\Psi$ is called an **implicitly-defined policy** for the domain $\mathcal{D}$. We say that $\Psi$ **implicitly defines** the policy $\Pi$ if and only if $\Pi = \bigcup_{(\Sigma, a) \in \Psi} \{(s, a) \mid s \in \mathcal{S}(\mathcal{V}), \Sigma \subseteq s\}$.

## Non-Deterministic Action Reversibility

Action reversibility, concerning the problem of whether the effects of an action can be undone by means of other actions, has been studied in deterministic settings (see Eiter, Erdem, and Faber (2008); Daum et al. (2016); Morak et al. (2020)).

In our work, we introduce the concept of action reversibility for non-deterministic actions in the FOND planning formalism. Non-deterministic action reversibility, in contrast to the deterministic variant, has to deal with non-deterministic actions—not only those whose effects we try to undo, but also those that have to be considered in the "reverting" process. In FOND planning, we usually consider two types of solutions, *weak* and *strong* (Cimatti et al. 2003). For a weak solution, we have a chance to achieve the goal if everything "goes well", while for a strong solution, there is a guarantee that the goal will eventually be achieved. We adopt these notions to establish *weak and strong action reversibility* such that for weak reversibility we require that there is a chance to undo all action effects, while for strong reversibility this has to be guaranteed. Note that in this paper we consider strong cyclic solutions that guarantee success if each effect of each non-deterministic action has a nonzero chance to oc-

cur if the action is applied (in other words, we consider the *fairness assumption*) (Aminof, Giacomo, and Rubin 2020).

Inspired by the work of Morak et al. (2020), we classify the notion of (non-deterministic) action reversibility into three categories, *S-reversibility*, *uniform reversibility* and *universal reversibility*, which will be now introduced.

## *S*-reversibility

Naturally, it might be useful to study whether an action $a$ is reversible in some subset of states $S \subseteq \mathcal{S}(\mathcal{V})$. That is, for each state from $S$ in which $a$ is applicable, we have to find a sequence of other actions that undoes the effects of $a$. This property is called *S-reversibility* (Morak et al. 2020).

To address non-deterministic actions, we introduce the notions of *weak S-reversibility* and *strong S-reversibility*, in the context of work of Cimatti et al. (2003) as described above. Informally speaking, we call an action *weakly S-reversible* if, in each state from $S$ in which the action is applicable, the action's effects can be undone by a sequence of (non-deterministic) actions while assuming the "correct" action effect always occurs. We call an action $a$ *strongly S-reversible* if, in each state $s \in S$ where $a$ is applicable, the action's effects are eventually undone by some policy $\Pi_s$ having precisely $s$ as its terminal state.

We also establish the notion of *action S-irreversibility*. It refers to a situation in which there is no way to undo any of the action's effects (even by means of a weak solution). The following definition formalises the above notions.

**Definition 1.** *Let $\mathcal{D} = \langle \mathcal{V}, \mathcal{A} \rangle$ be a planning domain, $a \in \mathcal{A}$ be an action, and $S \subseteq \mathcal{S}(\mathcal{V})$ be a set of states of the domain $\mathcal{D}$. The action $a$ is called **weakly (resp. strongly) S-reversible** in the domain $\mathcal{D}$ if and only if for each state $s \in S$, $pre(a) \subseteq s$, there exists a policy $\Pi$ for the domain $\mathcal{D}$ such that for each $s' \in \delta(s,a)$ it holds that $s \in \tau(\Pi, s')$ (resp. $\{s\} = \tau(\Pi, s')$ and for each state $s''$ that is reachable by $\Pi$ from the state $s'$ it holds that $s$ is reachable by $\Pi$ from $s''$). The action $a$ is called **S-irreversible** in the domain $\mathcal{D}$ if and only if for each state $s \in S$, $pre(a) \subseteq s$, there does not exist a policy $\Pi$ for the domain $\mathcal{D}$ such that for each $s' \in \delta(s,a)$ it holds that $s \in \tau(\Pi, s')$.*

**Example 1.** *Let $\mathcal{D}$ be a planning domain with two variables $door$ and $window$. Let $door$ have domain $\{open, closed\}$ and $window$ have domain $\{open, closed, broken\}$.*

*Let $vent$ be a deterministic action with $\{(window, closed)\}$ as its precondition, and $\{(door, open), (window, open)\}$ as effects. Furthermore, there are two additional non-deterministic actions: $close\text{-}door$ and $close\text{-}window$ with preconditions of a relevant entity being open and $eff(close\text{-}window) = \{\{(window, closed)\}, \{(window, broken)\}\}$ and $eff(close\text{-}door) = \{\{(door, closed)\}, \emptyset\}$.*

*The policy that applies $close\text{-}door$ in a state where both the door and the window are open and $close\text{-}window$ in a state where the window is open and the door is closed is a weak $\{\{(window, closed), (door, closed)\}\}$-reverse policy for the action $vent$. Similarly, there is a weak $\{\{(window, closed), (door, open)\}\}$-reverse policy for the action $vent$ that applies $close\text{-}window$ in a state where both the door and the window are open. Therefore, the action $vent$ is weakly $\{\{(window, closed), (door, closed)\}, \{(window, closed), (door, open)\}\}$-reversible.*

*As the application of both policies may eventually end in a broken window, neither of them is a strong reverse policy for the above set of states. Intuitively, $close\text{-}window$ action is irreversible in any state in which the action is applicable since no action can "fix" the broken window.*

*Now, consider the same domain, except that there are two variants of the former $close\text{-}window$. First, $close\text{-}window\text{-}draught\text{-}free$, which cannot break the window, and on top of the former action requires that the door is closed; and second, $close\text{-}window$, on top of the former action requires that the door is open, with the same effects as the former. In such a domain, the policy that applies $close\text{-}door$ if the door is not closed and $close\text{-}window\text{-}draught\text{-}free$ if the window is not closed (and the door is) is a strong $\{\{(window, closed), (door, closed)\}\}$-reverse policy for the action $vent$.*

## Uniform Reversibility

A special case of $S$-reversibility is *uniform reversibility*, which refers to the existence of a common solution concept that can undo the effects of the action in question, for each state of $S$ in which the action can be applied (Morak et al. 2020), i.e. the same solution concept reverses the effects of the action in question applied in any state from the set $S$ where the action is applicable. The following definition formalises the notions of weak and strong uniform $S$-reversibility for non-deterministic actions.

**Definition 2.** *Let $\mathcal{D} = \langle \mathcal{V}, \mathcal{A} \rangle$ be a planning domain, $a \in \mathcal{A}$ be an action, $S \subseteq \mathcal{S}(\mathcal{V})$ be a set of states of the domain $\mathcal{D}$. The action $a$ is called **weakly (resp. strongly) uniformly S-reversible** in the domain $\mathcal{D}$ if and only if exists policy $\Pi$ for the domain $\mathcal{D}$ such that for each state $s \in S$, $pre(a) \subseteq s$, it holds that for each $s' \in \delta(s,a)$ we have $s \in \tau(\Pi, s')$ (resp. $\{s\} = \tau(\Pi, s')$ and for each state $s''$ that is reachable by $\Pi$ from the state $s'$ it holds that $s$ is reachable by $\Pi$ from $s''$).*

Having a single policy that can undo the effects of an action in all relevant situations (i.e. states from $S$ in which $a$ is applicable) is practical as we do not need to consider multiple state-specific policies. Analogously to the notion of *reverse plan* in classical planning (Morak et al. 2020), we define the notion of *reverse policy*.

**Definition 3.** *Let $\mathcal{D} = \langle \mathcal{V}, \mathcal{A} \rangle$ be a planning domain and $a \in \mathcal{A}$ be an action. A policy $\Pi$ for the domain $\mathcal{D}$ is called a **weak (resp. strong) S-reverse policy** for the action $a$ if and only if the action $a$ is weakly (resp. strongly) uniformly S-reversible by the policy $\Pi$.*

**Example 2.** *Consider the domain of Example 1. Policy which applies $door\text{-}close$ if the door is open and $close\text{-}window$ if the window is open is a weak uniform $\{\{(window, closed), (door, closed)\}, \{(window, closed), (door, open)\}\}$-reverse policy (however, the policy relates some states with multiple actions).*

## Universal Reversibility

*Universal reversibility* is a specific case of *S-reversibility* covering all states, i.e. $S = \mathcal{S}(\mathcal{V})$. Informally speaking, if

an action is universally reversible (or irreversible), its effects can always (or never) be undone.

**Definition 4.** *Let $\mathcal{D} = \langle \mathcal{V}, \mathcal{A} \rangle$ be a planning domain and $a \in \mathcal{A}$ be an action. The action $a$ is called **weakly (resp. strongly) universally (uniformly) reversible** in the domain $\mathcal{D}$ if and only if the action $a$ is weakly (resp. strongly) (uniformly) $\mathcal{S}(\mathcal{V})$-reversible in the domain $\mathcal{D}$. The action $a$ is called **universally irreversible** in the domain $\mathcal{D}$ if and only if the action $a$ is $\mathcal{S}(\mathcal{V})$-irreversible in the domain $\mathcal{D}$.*

**Example 3.** *Consider the domain of Example 1. The states with a closed window and the door being open or closed are the only states where the action $vent$ is applicable. Recall that there is a weak reverse policy for any such state. Therefore, the action $vent$ is weakly universally reversible. The common policy of Example 2 proves the weak universal uniform reversibility of the action $vent$.*

**Example 4.** *Now, as an example of strong universal uniform reversibility, let us consider a domain of well-known Transport domain (Helmert, Do, and Refanidis 2010). In the domain, packages have to be transported from one location to another through various vehicles. Packages can be loaded and unloaded into vehicles without restrictions and the loading or unloading process can fail, leaving the package in its former position (at the location or inside the truck, respectively). The actions of the $load$ and $unload$ "families" are strongly universally uniformly reversible as if the package is successfully loaded or unloaded, respectively, we can always undo it by unloading, or loading the package back.*

## Theoretical Properties

Having defined the relevant notions, we will first establish some theoretical properties that follow from our definitions.

It can be easily derived that weak or strong (uniform) $S$-(ir)reversibility implies weak or strong (uniform) $S'$-(ir)reversibility if $S' \subseteq S$. Weak or strong uniform $S$-reversibility implies weak or strong $S$-reversibility, and strong (uniform) $S$-reversibility implies weak (uniform) $S$-reversibility. Furthermore, we can observe that if an action is weakly (resp. strongly) $S_1$- and $S_2$-reversible, then it is weakly (resp. strongly) $(S_1 \cup S_2)$-reversible. However, for uniform reversibility such an implication does not hold in general.

A more interesting property of any $S$-reverse policy is that it cannot assign any action to any state of $S$ in which the action the policy is reverting is applicable.

**Example 5.** *Consider a domain that describes the result of coin tossing with the only variable with domain $\{heads, tails\}$ and $toss\text{-}coin$ (without preconditions). The action has a strong universal uniform $\{(coin, head)\}$-reverse policy that applies $toss\text{-}coin$ as long as the state is $\{(coin, tails)\}$. The same holds symmetrically for $tails$. Hence, the action is weakly and strongly universally reversible. However, there is no weak or strong universal reverse policy, since, e.g., the policy of union of mentioned policies has no terminal state (it relates each state with some action).*

**Proposition 6.** *Let $\mathcal{D} = \langle \mathcal{V}, \mathcal{A} \rangle$ be a planning domain, $\Pi$ be a policy for the domain $\mathcal{D}$, and $a \in \mathcal{A}$ be an action. If $\Pi$ is a weak (resp. strong) $S$-reverse policy for the action $a$, then $\{s \mid s \in S, pre(a) \subseteq s\} \cap \sigma(\Pi) = \emptyset$.*

*Proof.* Proof by a contradiction. Assume that $\{s' \mid s' \in S, pre(a) \subseteq s'\} \cap \sigma(\Pi) \neq \emptyset$. Then, there is a state $s \in \{s' \mid s' \in S, pre(a) \subseteq s'\} \cap \sigma(\Pi)$. According to the assumption and since $\{s\} \subseteq S$, the action $a$ is weakly (resp. strongly) uniformly $\{s\}$-reversible by the policy $\Pi$. Since $s$ is also in $\sigma(\Pi)$, $s$ cannot be a terminal state of the policy $\Pi$. Therefore, for each $s' \in \delta(s, a)$ it holds that $s \notin \tau(\Pi, s')$. This is in contradiction with the assumption of $\Pi$ being a weak (resp. strong) $S$-reverse policy for the action $a$. $\qquad\square$

An important consequence of Proposition 6 concerns universal uniform reversibility. It simply follows that when the set $S$ is equal to the set $\mathcal{S}(\mathcal{V})$, then the situation of Proposition 6 can be simplified to the fact that the universal reverse policy can contain only the states in which the action we want to reverse is not applicable.

In contrast to deterministic planning which deals with action sequences, policies have a different structure as they assign appropriate actions to apply in relevant states. This property gives grounds to raise a question of how combining two policies affects $S$-reversibility of non-deterministic actions. In the case of weak reversibility, in which reverse policies are constructed along a sequence of actions, we can safely combine policies as long as the other policy does not compromise the assumption of Proposition 6.

**Theorem 7.** *Let $\mathcal{D} = \langle \mathcal{V}, \mathcal{A} \rangle$ be a planning domain, $a \in \mathcal{A}$ be an action, $\Pi_1$ be a weak $S$-reverse policy for the action $a$ and $\Pi_2$ be a policy for the domain $\mathcal{D}$. The policy $\Pi_1 \cup \Pi_2$ is a weak $S$-reverse policy for the action $a$ if and only if $\{s \mid s \in S, pre(a) \subseteq s\} \cap \sigma(\Pi_2) = \emptyset$.*

*Proof.* The "if" part is proven by a contraposition. Let us assume that there is some state $s' \in \{s \mid s \in S, pre(a) \subseteq s\} \cap \sigma(\Pi_2)$. Then, as $\sigma(\Pi_2) \subseteq \sigma(\Pi_1 \cup \Pi_2)$, $s' \in \{s \mid s \in S, pre(a) \subseteq s\} \cap \sigma(\Pi_1 \cup \Pi_2)$. That violates the assumption of Proposition 6 and hence the policy $\Pi_1 \cup \Pi_2$ cannot be a weak $S$-reversible policy for the action $a$.

For the "only if" part, we can observe that extending a policy (by adding a pair of some state and some action) does not affect the reachability of any state that was reachable before (including the terminal states). From the assumption $\{s \mid s \in S, pre(a) \subseteq s\} \cap \sigma(\Pi_2) = \emptyset$ and the fact that $\{s \mid s \in S, pre(a) \subseteq s\} \cap \sigma(\Pi_1) = \emptyset$ (see Proposition 6), we get $\{s \mid s \in S, pre(a) \subseteq s\} \cap \sigma(\Pi_1 \cup \Pi_2) = \emptyset$. Therefore, all states $\{s \mid s \in S, pre(a) \subseteq s\}$ are terminal with respect to a given starting state for the policy $\Pi_1 \cup \Pi_2$ (see the definition of terminal states). Hence, the policy $\Pi_1 \cup \Pi_2$ is a weak $S$-reverse policy for the action $a$. $\qquad\square$

Regarding the strong $S$-reverse policy, since every strong $S$-reverse policy is also a weak $S$-reverse policy, the same conditions apply in this case as well. However, the conditions of Theorem 7 are not sufficient since, if the second policy assigns an action to a state that is considered by the first strong $S$-reverse policy, the union of such policies could introduce a different terminal state; and this would violate the definition of strong $S$-reverse policy. Hence, to combine a strong $S$-reverse policy with a second one, the latter should not interfere with the former, as summarised in the following theorem.

**Theorem 8.** *Let $\mathcal{D} = \langle \mathcal{V}, \mathcal{A} \rangle$ be a planning domain, $a \in \mathcal{A}$ be an action, $\Pi_1$ be a strong $S$-reverse policy for the action $a$ and $\Pi_2$ be a policy for the domain $\mathcal{D}$. If $\sigma(\Pi_2) \cap \{s \mid s \in S, pre(a) \subseteq s\} = \emptyset$ and $\sigma(\Pi_1) \cap \sigma(\Pi_2) = \emptyset$, then $\Pi_1 \cup \Pi_2$ is a strong $S$-reverse policy for the action $a$.*

*Proof.* Let $s$ be a state $s \in \{s'' \mid s'' \in S, pre(a) \subseteq s''\}$. From the assumption of $\Pi_1$ being a strong $S$-reverse policy for the action $a$, we get $\{s\} = \tau(\Pi_1, s')$ for all $s' \in \delta(s, a)$. Let $D^0$ be a set of reachable states from the state $s'$ by policy $\Pi_1$. The set can be divided into two disjoint sets $D^0 \cap \sigma(\Pi_1)$ and $D^0 \setminus \sigma(\Pi_1) = \tau(\Pi_1, s') = \{s\}$.

Now, we show that if we add an arbitrary state-action pair $(s'', a'') \in \Pi_2$ to the policy $\Pi_1$, the set of reachable states $D^0$ from the state $s'$ by the extended policy remains unchanged. Therefore, if the policy $\Pi_1$ is extended by any state-action pair $(s'', a'') \in \Pi_2$, i.e., $\Pi_1' = \Pi_1 \cup \{(s'', a'')\}$, $D^0$ remains unchanged, because $s'' \notin D^0 \cap \sigma(\Pi_1')$ (due to the fact that $\sigma(\Pi_1) \cap \sigma(\Pi_2) = \emptyset$) and neither is in $D^0 \setminus \sigma(\Pi_1') = \tau(\Pi_1, s') = \{s\}$ (due to the fact that $\sigma(\Pi_2) \cap \{s''' \mid s''' \in S, pre(a) \subseteq s'''\} = \emptyset$).

Therefore, we can derive that $D^0$ remains the same for the combined policy $\Pi_1 \cup \Pi_2$. Furthermore, we can see that no other state could become terminal since no other state became reachable, i.e., $\tau(\Pi_1 \cup \Pi_2, s') \subseteq \{s\}$. From Theorem 7 we know $s \in \tau(\Pi_1 \cup \Pi_2, s')$. Hence, $\Pi_1 \cup \Pi_2$ is a strong $S$-reverse policy for the action $a$. $\qquad\square$

A practical consequence of the above theorems is that they provide conditions under which we can merge two reverse policies into one. If $\Pi_1$ is a weak (resp. strong) $S_1$-reverse policy for some action $a$ and $\Pi_2$ is a weak (resp. strong) $S_2$-reverse policy for the action $a$, $\Pi_1 \cup \Pi_2$ is a weak (resp. strong) $(S_1 \cup S_2)$-reverse policy for $a$ if the conditions of Theorem 7 (resp. Theorem 8) are satisfied. We note that the possibility of combining policies is not applicable for the deterministic case in which reverse plans are considered sequences of actions (Eiter, Erdem, and Faber 2008; Morak et al. 2020).

## Determining Universal Uniform Reversibility

To determine universal uniform reversibility as well as universal irreversibility, we took inspiration from the work of Chrpa, Faber, and Morak (2021) who studied such cases of reversibility in the deterministic context. In summary, to determine universal uniform reversibility (for deterministic actions), it is sufficient to consider only actions that contain only variables present in the precondition of a "to be reversed" action. Such a property is practically very useful, as it usually (considerably) simplifies the problem.

However, in the non-deterministic context, the theoretical findings of Chrpa, Faber, and Morak (2021) can only be partially adopted. Due to the different nature of policies, we show that allowing only actions operating over the set of variables present in the precondition of a "to be reversed" action while looking for weak or strong universal reverse policies is a sufficient but not necessary condition. In other words, if a strong or weak reverse policy containing only "restricted" actions is found, then such a policy is universal,

but the opposite implication does not generally hold (as we argue later in this section).

## Theoretical Properties

At first, we show that for an action to be strongly universally uniformly reversible, its effects may modify only variables that are also present in its preconditions. Note that an analogous claim also holds for the deterministic case (Chrpa, Faber, and Morak 2021). The intuition behind the claim is that if a variable is present in action's effects but not present in its precondition, then multiple states (each refers to a different value of the variable while assuming that the variable can have at least two different values) "collapse" into a single state referring to the value of the variable present in the action's effects.

**Lemma 9.** *Let $\mathcal{D} = \langle \mathcal{V}, \mathcal{A} \rangle$ be a planning domain such that $\forall v \in \mathcal{V} : |dom(v)| \geq 2$ and $a \in \mathcal{A}$ be an action. If $\bigcup_{e \in \mathit{eff}(a)} vars(e) \nsubseteq vars(pre(a))$, then the action $a$ is not strongly universally uniformly reversible.*

*Proof.* Assume there exists a strong universal reverse policy $\Pi$ for the action $a$.

W.l.o.g., let $e \in \mathit{eff}(a)$ be an action effect for which there exists $v \in vars(e) \setminus vars(pre(a))$. Since $|dom(v)| \geq 2$, there exist two distinct states $s_1$, $s_2$ in which $a$ is applicable and which differ in the value of the variable $v$. As states $s_1$ and $s_2$ differ only in the value of the variable $v$ and since the effect $e$ modifies $v$, we have $s' = \gamma(s_1, a_e^d) = \gamma(s_2, a_e^d)$.

As we initially assumed, $\Pi$ is a strong universal reverse policy for the action $a$ and this implies $\tau(\Pi, \gamma(s_1, a_e^d)) = \{s_1\}$ and $\tau(\Pi, \gamma(s_2, a_e^d)) = \{s_2\}$ (see Definition 2). However, since $s' = \gamma(s_1, a_e^d) = \gamma(s_2, a_e^d)$, we get $\tau(\Pi, s') = \{s_1, s_2\}$. This is in contradiction with $s_1 \neq s_2$. $\qquad\square$

In the case of weak universal uniform reversibility, an analogous claim cannot be made. For a weak universal reverse policy $\Pi'$, we can derive an analogous claim as we did in the proof of Lemma 9, that is, $\tau(\Pi', s') = \{s_1, s_2\}$. Such a claim does not contradict the definition of weak uniform reversibility, as both cases, that is, $s_1 \in \tau(\Pi', s')$ and $s_2 \in \tau(\Pi', s')$, might be satisfied.

As an example, consider a bowling domain, with a variable $bowled$ with the domain $\{true, false\}$ and an action that resets bowling pins that restores any combination of pins being down or standing to all pins standing, and that sets $bowled$ to $false$. The action is applicable if and only $bowled$ is $true$. The action's effects contain variables of pins that are not in its preconditions. There is another action that rolls the bowling ball, without any preconditions, and which can non-deterministically push over none, some, or all standing pins, and always sets $bowled$ to $true$. The rolling action can weakly universally uniformly undo the reset action, as each state before the reset can be obtained, but it cannot strongly universally uniformly undo the reset action, as no state before the reset can be guaranteed to be obtained.

The following theorem shows that if we generate a weak or strong universal reverse policy in a restricted state and action space that takes into account only variables that are

present in the precondition of "to be reversed" action, then such a reverse policy is universal.

**Theorem 10.** *Let $\mathcal{D} = \langle \mathcal{V}, \mathcal{A} \rangle$ be a planning domain, $a \in \mathcal{A}$ be an action such that $\bigcup_{e \in \mathit{eff}(a)} \mathit{vars}(e) \subseteq \mathit{vars}(\mathit{pre}(a))$. If exists an implicitly defined policy $\Psi$ for the domain $\mathcal{D}$ such that $\forall (\Sigma', a') \in \Psi : \mathit{vars}(\Sigma') \subseteq \mathit{vars}(\mathit{pre}(a))$ and that the policy $\Pi$ implicitly defined by $\Psi$ is weak (resp. strong) $S$-reverse policy for the action $a$ such that $\{s \mid s \in S, \mathit{pre}(a) \subseteq s\} \neq \emptyset$ and $\forall (s', a') \in \Pi : \mathit{vars}(a') \subseteq \mathit{vars}(\mathit{pre}(a))$, then $\Pi$ is a weak (resp. strong) universal reverse policy for the action $a$.*

*Proof.* Let $s \in S$ be a state such that $\mathit{pre}(a) \subseteq s$ and $\Pi$ be a weak (resp. strong) $\{s\}$-reverse policy for the action $a$ implicitly defined by $\Psi$.

To show that $\Pi$ is a weak (resp. strong) universal reverse policy for the action $a$, we need to prove that for each state $s' \in \mathcal{S}(\mathcal{V})$ such that $\mathit{pre}(a) \subseteq s'$ the policy $\Pi$ is also a weak (resp. strong) $\{s'\}$-reverse policy for the action $a$. If there does not exist any other state $s' \in \mathcal{S}(\mathcal{V})$ such that $\mathit{pre}(a) \subseteq s'$ and $s \neq s'$, then the policy $\Pi$ is a weak (resp. strong) universal reverse policy for the action $a$. If there exists such a state $s'$, then we can observe that $s'$ differs from $s$ only in values of variables that are not part of $\mathit{vars}(\mathit{pre}(a))$ (otherwise it would compromise the applicability of $a$).

Assumption $\bigcup_{e \in \mathit{eff}(a)} \mathit{vars}(e) \subseteq \mathit{vars}(\mathit{pre}(a))$ states that any effect $e$ of the action $a$ can modify only the variables present in $\mathit{pre}(a)$. Therefore, the values of the variables of $\mathcal{V} \setminus \mathit{vars}(\mathit{pre}(a))$ remain unchanged after the application of the action $a$ regardless of what effect takes place.

As for any state-action pair $(s_x, a_x) \in \Pi$ it is the case that $\mathit{vars}(a_x) \subseteq \mathit{vars}(\mathit{pre}(a))$, none of the actions of the policy $\Pi$ can change any variable from $\mathcal{V} \setminus \mathit{vars}(\mathit{pre}(a))$ or require a specific value of any of such variables.

From the assumption stating $\forall (\Sigma', a') \in \Psi : \mathit{vars}(\Sigma') \subseteq \mathit{vars}(\mathit{pre}(a))$ it holds that each variable assignment $\Sigma'$ does not contain variables from $\mathcal{V} \setminus \mathit{vars}(\mathit{pre}(a))$. Therefore, the policy $\Pi$ behaves independently on values of variables $\mathcal{V} \setminus \mathit{vars}(\mathit{pre}(a))$ which means that $\Pi$ yields the same outcome (apart from the values of the variables of $\mathcal{V} \setminus \mathit{vars}(\mathit{pre}(a))$ that remain constant) from both states $s$ and $s'$.

According to the other theorem assumption, if we apply $\Pi$ in any state in $\delta(s, a)$, we may (resp. have to) return to the state $s$, which is terminal. As we have shown that $\Pi$ and $a$ can modify only variables present in $\mathit{pre}(a)$, we can derive that for each state $s' \in \mathcal{S}(\mathcal{V})$ in which $a$ is applicable, $\Pi$ is weak (resp. strong) $\{s'\}$-reverse policy for $a$. Hence, the policy $\Pi$ is a weak (resp. strong) universal reverse policy for the action $a$. $\qquad\square$

In contrast to deterministic universal uniform reversibility (Chrpa, Faber, and Morak 2021), the opposite implication, i.e. there is a weak/strong universal reverse policy for an action if Theorem 10's assumption is true, does not hold.

Regarding weak universal uniform reversibility, Lemma 9 indicates that a "to be reversed" action might modify variables that are not present in its precondition and still be (potentially) weakly universally uniformly reversible. This observation can also be applied to other actions that are part of the weak universal reverse policy. As an example, we might have an action whose non-deterministic effects contain all states in which the "to be reversed" action is applicable.

In the strong universal uniform reversibility case, the issue with the opposite implication of Theorem 10 is in the possibility of merging policies. If the conditions of Theorem 8 are satisfied, we can merge two strong reverse policies to obtain another strong reverse policy applicable to the union of states of the individual policies. Hence, it is possible to obtain a strong universal reverse policy by combining more specific strong reverse policies.

As an example, let us have a domain where an agent can move between three locations – $A$, $B$, $C$. However, there is another variable that determines whether the agent can move from $B$ to $A$, or from $B$ to $C$. Moving from $A$ to $B$ and from $C$ to $A$ is unrestricted. Depending on the value of the other variable, we undo the effects of moving from $A$ to $B$ by either moving directly back to $A$, or moving through $C$. These two reverse policies can be combined, as the states they are operating over are disjoint (because of different values of the other variable). The combined policy is a strong universal reverse policy for the action that moves us from $A$ to $B$. As the other variable is involved in the policy and there is no other strong universal reverse policy, there is no $\Psi$ of Theorem 10.

## Compilation

Theorem 10 provides us with a blueprint on how classical (respectively, FOND planning) can be leveraged in finding weak (resp. strong) universal reverse policies.

The following theorem shows how a weak universal reverse policy can be generated by means of classical planning by specifying multiple classical planning tasks (one for each determinisation of the "to be reversed" action). Since, for weak reversibility, we need only to consider the "correct" effects of actions, we replace each non-deterministic action with its respective determinisations. We consider only determinisations that operate only on the variables present in the precondition of the "to be reversed" action.

**Theorem 11.** *Let $\mathcal{D} = \langle \mathcal{V}, \mathcal{A} \rangle$ be a planning domain and $a \in \mathcal{A}$ be an action such that $\bigcup_{e \in \mathit{eff}(a)} \mathit{vars}(e) \subseteq \mathit{vars}(\mathit{pre}(a))$. If for each determinisation $a_e^d$ of the action $a$ a plan $\pi_e$ is a goal plan for the SAS$^+$ planning task $\mathcal{T}_e^d = \langle \langle \mathit{vars}(\mathit{pre}(a)), \{(a')_e^d \mid a' \in \mathcal{A}, e \in \mathit{eff}(e), \mathit{vars}((a')_e^d) \subseteq \mathit{vars}(\mathit{pre}(a))\}\rangle, \mathit{ha}(a_e^d), \mathit{pre}(a)\rangle$, then the action $a$ is weakly universally uniformly reversible.*

*Proof (Sketch).* Each plan $\pi_e = \langle a_{e_1}, \ldots, a_{e_n} \rangle$ can be transformed into a policy $\Pi_e = \{(\gamma(\mathit{ha}(a_e^d), \langle a_{e_1}, \ldots, a_{e_{i-1}} \rangle), a_{e_i}) \mid i \in \mathbb{N}, 1 \leq i \leq n\}$. Since for each effect $e_i$ determined by the corresponding determinisation $a_{e_i}$ it is the case that $\mathit{vars}(e_i) \subseteq \mathit{vars}(\mathit{pre}(a))$, we can observe that none of the variables other than $\mathit{vars}(\mathit{pre}(a))$ is modified in the process. Also, each determinisation $a_e^d$ of the action $a$ modifies only the variables of $\mathit{vars}(\mathit{pre}(a))$. All policies resulting from transformed plans that are solutions of the SAS$^+$ tasks for each determinisation $a_e^d$ contain $\mathit{pre}(a)$ as one of the terminal states. By combining all the policies into one (meeting the conditions of Theorem 7) we ob-

tain an implicitly-defined policy for the domain $\langle \mathcal{V}, \{(a')_e^d \mid a' \in \mathcal{A}, e \in \mathit{eff}(e), \mathit{vars}((a')_e^d)\} \cup \{a\}\rangle$. If we replace the determinisations by their corresponding stochastic actions of $\mathcal{A}$, we obtain an equivalent (from the perspective of weak reversibility) implicitly-defined policy $\Psi$ for the domain $\langle \mathcal{V}, \mathcal{A}\rangle$. Then, a policy implicitly defined by $\Psi$ is a weak universal reverse policy for the action $a$. $\square$

A similar claim can be made about strong universal uniform reversibility, albeit in the context of FOND planning, since strong universal uniform reversibility needs to guarantee that the only terminal state is the state in which a "to be reversed" action is applied. The compilation of the problem of finding a strong universal reverse policy to a FOND planning task is analogous to the weak universal uniform reversibility case. In addition, to be able to merge the found policies, their sets of related states have to be pairwise disjoint, as Theorem 8 suggests.

**Theorem 12.** *Let* $\mathcal{D} = \langle \mathcal{V}, \mathcal{A}\rangle$ *be a planning domain and* $a \in \mathcal{A}$ *be an action such that* $\bigcup_{e \in \mathit{eff}(a)} \mathit{vars}(e) \subseteq \mathit{vars}(\mathit{pre}(a))$. *If for each determinisation* $a_e^d$ *there exists a strong goal policy* $\Pi_e$ *for the FOND planning task* $\mathcal{T} = \langle\langle \mathit{vars}(\mathit{pre}(a)), \{a' \mid a' \in \mathcal{A}, \mathit{vars}(a') \subseteq \mathit{vars}(\mathit{pre}(a))\}\rangle, \mathit{ha}(a_e^d), \mathit{pre}(a)\rangle$ *such that the sets* $\sigma(\Pi_e)$ *are pairwise disjoint, then the action* $a$ *is strongly universally uniformly reversible.*

*Proof (Sketch).* It can be observed that each policy $\Pi_e$ is an implicitly defined policy that strongly universally uniformly reverses the determinisation $a_e^d$. These policies can be combined (since the sets of related states are pairwise disjoint) according to Theorem 8. Hence, the resulting policy implicitly defined by $\bigcup_{e \in \mathit{eff}(a)} \Pi_e$ is a strong universal reverse policy for the action $a$. $\square$

We also consider the possibility of identifying universal irreversibility. The idea is derived from the work of Chrpa, Faber, and Morak (2021) that concerns universal irreversibility of deterministic actions. Informally speaking, if for some effect of a non-deterministic action, we cannot reachieve the precondition of that action, then the action is universally irreversible. We can "project" the problem onto the variables related to a "to be reversed" action $a$, i.e., $\mathit{vars}(a)$. In contrast to the above theorems, we do not discard actions that also operate on the other variables, but we project their preconditions and effects to $\mathit{vars}(a)$.

Let $\Sigma$ be a variable assignment over the set of variables $\mathcal{V}$ and let $\mathcal{V}' \subseteq \mathcal{V}$ be a set of variables. The projection of $\Sigma$ on the set of variables $\mathcal{V}'$ is a variable assignment $\Sigma_{|\mathcal{V}'} = \{(v, x) \in \Sigma \mid v \in \mathcal{V}'\}$. Furthermore, the projection of an action $a$ on the set of variables $\mathcal{V}'$ is an action $a_{|\mathcal{V}'} = (\mathit{pre}(a)_{|\mathcal{V}'}, \{e_{|\mathcal{V}'} \mid e \in \mathit{eff}(a)\})$.

**Theorem 13.** *Let* $\mathcal{D} = \langle \mathcal{V}, \mathcal{A}\rangle$ *be a planning domain and* $a \in \mathcal{A}$ *be an action. If for any determinisation* $a_e^d$ *of the action* $a$ *the SAS$^+$ planning task* $((\mathit{vars}(a), \{((a')_{e'}^d)_{|\mathit{vars}(a)} \mid a' \in \mathcal{A}, e' \in \mathit{eff}(a')\}), \mathit{ha}(a_e^d), \mathit{pre}(a_e^d))$ *is unsolvable, then the action* $a$ *is universally irreversible.*

| | | $\not\subseteq (\neg S)$ | | $\subseteq$ | | | | |
|---|---|---|---|---|---|---|---|---|
| Dom. | $\|\mathcal{A}\|$ | I | ? | W | FM | S | ? | I |
| A. | 5 | 0 | 0 | 4 | 1 | 3 | 1 | 0 |
| BE. W. | 7 | 0 | 0 | 6 | 2 | 4 | 0 | 1 |
| BL. W. | 190 | 0 | 185 | 5 | 0 | 5 | 0 | 0 |
| B. F. | 5 | 0 | 0 | 2 | 0 | 2 | 0 | 3 |
| C. | 3 | 0 | 0 | 0 | 0 | 0 | 0 | 3 |
| D. | 5 | 4 | 1 | 0 | 0 | 0 | 0 | 0 |
| E. O. | 27 | 0 | 0 | 21 | 0 | 21 | 0 | 6 |
| EL. | 41 | 3 | 30 | 8 | 0 | 8 | 0 | 0 |
| E. BL. | 85 | 35 | 45 | 0 | 0 | 0 | 5 | 0 |
| FA. | 51 | 25 | 26 | 0 | 0 | 0 | 0 | 0 |
| F. R. | 46 | 6 | 4 | 22 | 0 | 22 | 8 | 6 |
| FO. | 150 | 7 | 119 | 4 | 0 | 4 | 10 | 10 |
| I. | 24 | 0 | 0 | 20 | 0 | 20 | 0 | 4 |
| M. | 179 | 0 | 3 | 158 | 0 | 158 | 0 | 18 |
| R. | 3 | 2 | 0 | 0 | 0 | 0 | 0 | 1 |
| S. TI. | 211 | 3 | 78 | 124 | 0 | 124 | 6 | 0 |
| TI. | 52 | 7 | 1 | 0 | 0 | 0 | 44 | 0 |
| TI. T. | 24 | 0 | 10 | 10 | 0 | 10 | 0 | 4 |
| T. TI. | 11 | 3 | 0 | 0 | 0 | 0 | 0 | 8 |
| Z. | 740 | 0 | 96 | 504 | 0 | 504 | 140 | 0 |

Table 1: Results of identification of action reversibility. The table headers depict proven action properties: number of actions in the domain ($|\mathcal{A}|$), weak (resp. strong) universal uniform reversibility (W) (resp. S), universal irreversibility (I), whether $\bigcup_{e \in \mathit{eff}(a)} \mathit{vars}(e) \subseteq \mathit{vars}(\mathit{pre}(a))$ ($\subseteq$ or $\not\subseteq$), and "?" indicates "no class of action (ir)reversibility was identified". The column labelled as "FM" represents a proven weak universal uniform reversibility, where uniform strong reverse policies were found for all determinizations, but due to the condition on empty related state intersection, we were unable to merge them. "$\neg S$" stands for actions which are proven not to be strongly universally uniformly reversible.

*Proof.* If the specified SAS$^+$ planning task is unsolvable, it means that the precondition of $a$ is unreachable from some of its effects. Since we focus only on a subset of variables, the applicability of actions is more optimistic than in the general case. So, if an abstract task created by projecting into a subset of variables is unsolvable, then the original task is unsolvable as well (Helmert, Haslum, and Hoffmann 2007). Hence, we can derive that if for any determinisation of $a$, the specified SAS$^+$ planning task is unsolvable, then $a$ is universally irreversible. $\square$

## Experiments

The section presents empirical evidence on the existence of investigated phenomena in many benchmark domains. Based on the claims of previous sections, we have designed and performed the experiments for the resolution of investigated classes of non-deterministic action reversibility.

We have evaluated our approaches for non-deterministic action reversibility on 20 FOND domains of two sets of benchmarks: one from the repository of the PRP planner (Muise, McIlraith, and Beck 2012; Muise, Belle, and McIlraith 2014; Muise, McIlraith, and Belle 2014), and second set proposed by Geffner and Geffner (2018); viz., Acrobatics (A.), Beam Walk (BE. W.), Blocks World (BL. W.),

Bus Fare (B. F.), Climber (C.), Doors (D.), Earth Observation (E. O.), Elevators (EL.), Exploding Blocks World (E. BL.), Faults (FA.), First Responders (F. R.), Forest (FO.), Islands (I.), Miner (M.), River (R.), Spiky Tire World (S. TI.), Tire World (TI.), Tire World Truck (TI. TRU.), Triangle Tire World (TRI. TI.), and Zeno Travel (Z.).

For each action, we initially check whether they satisfy the condition $\bigcup_{e \in \mathit{eff}(a)} \mathit{vars}(e) \subseteq \mathit{vars}(\mathit{pre}(a))$. If the action does not satisfy the condition, we use Theorem 13 to check its universal irreversibility. Based on the result, we either conclude "universally irreversible" or "we have not identified anything, besides the action is not strongly uniformly universally reversible" (since the action does not satisfy the conditions, from Theorem 10 we know that the action is not strongly universally uniformly reversible; Theorem 11 is inapplicable and Theorem 13 is an implication only). If the action satisfies the initial condition, we check if it is weakly universally uniformly reversible (which is a necessary condition for strong reversibility) by leveraging Theorem 11. Based on the result, strong universal uniform reversibility or universal irreversibility is checked using the respective Theorems 12 and 13, respectively. If all of them "fail", we conclude that "we have not identified anything", which is denoted in Table 1 as "?". To check weak reversibility and irreversibility, we use the LAMA planner (Richter and Westphal 2010) and to check strong reversibility, we use the PRP planner (both planners are built on top of the Fast Downward planner framework (Helmert 2006)).

All experiments[1] ran on a machine with an Intel® Core™ i7-7700HQ processor and 32 GB of DDR4 RAM operating at a frequency of 2400 MHz. The operating system was Ubuntu 22.04.3 in WSL 2 (version 2.0.9.0) of Windows 10.

### Results

Table 1 provides an overview of what actions have been identified as (weakly or strongly) universally uniformly reversible, universally irreversible, or unidentified.

The results indicate that the weakly universally uniformly reversible actions identified by Theorem 11 are likely also identified as strongly universally uniformly reversible by Theorem 12, since this happened in the vast majority of situations in our experiments. The only exceptions are three actions for which we have found strong universal uniform reverse plans for each determinisation (see Theorem 12), but we were unable to merge them into a general policy since the reverse plans had conflicting states.

On average, we were able to identify some class of (ir)reversibility in approximately $68.30 \pm 33.64\%$ of actions, with the lowest relative amount of approx. $2.63\%$ in the BL. W. domain, while in the BE. W., B. F., C., E. O., I., R. and TRI. TI. domains all the actions were identified in some class. In domains such as FA., FO. or E. BL., most of the actions did not satisfy the "$\subseteq$" condition and hence could not be identified as weakly universally uniformly reversible through our theoretical study.

Overall, on average, approx. $60.91 \pm 40.13\%$ of actions satisfied the "$\subseteq$" condition on action's preconditions. In an

---

[1] We plan to release the code if the paper gets accepted.

"average" case, approx. $33.49\%$ of actions were identified as universally irreversible, $34.81\%$ as weakly universally uniformly reversible, $32.39\%$ as strongly universally uniformly reversible, and $31.70\%$ of actions remained undetermined (columns labeled with the question mark in Table 1) out of which $23.88\%$ is proven not to be strongly universally uniformly reversible (with respect to the total number of actions). The relatively high number of undetermined actions is caused by the requirement for universality and uniformity. Although these requirements are practically desirable, actions might not always conform to them. Also, as we have shown, the methods are not theoretically complete and hence some cases might not have been identified (as we observed in the case of universal irreversibility in TI. domain).

On average, the measured mean compilation time per action in a domain is $7.037 \pm 5.748$ milliseconds for weak, $7.028 \pm 5.772$ milliseconds for strong, and $9.113 \pm 5.715$ milliseconds for irreversibility reformulation with minimal, resp. maximum values, $0.994$, $0.846$, $3.031$ milliseconds, resp. $17.337$, $17.743$, $22.441$ milliseconds. As for runtimes for solving the reformulations, we have measured the time required to find a plan or policy, or to decide the (un)solvability of an abstract SAS$^+$ planning task for all determinisations. The mean runtimes are $156.111 \pm 60.408$, $47.712 \pm 18.495$ and $186.015 \pm 81.118$ milliseconds to determine weak universal uniform reversibility, strong universal uniform reversibility and universal irreversibility according to Theorems 11, 12 and 13, respectively. Minimum values are $118.091$, $22.970$, $120.439$, and maximum values are $462.132$, $87.549$, and $711.832$ milliseconds, respectively. The results show the practical viability of our methods.

### Conclusion

In this paper, we have conceptualised the notions of *weak and strong reversibility* of non-deterministic actions in FOND planning. These notions are inspired by weak and strong plans in FOND planning (Cimatti et al. 2003) and share the same meaning, i.e., weak reversibility refers to a possibility of undoing all effects of an action, while strong reversibility refers to the certainty of undoing all effects. We specifically focused on *universal uniform* cases that refer to the fact that effects of a non-deterministic action can always be undone by the same reverse policy. We proposed methods based on compiling the weak and strong universal uniform reversibility problem into classical or FOND planning, respectively, and we proposed a method for determining universal irreversibility via classical planning.

An experimental evaluation that we conducted on existing FOND benchmarks has shown that we were able to identify a type of reversibility for about $56.32\%$ actions and the running time of any of the methods was in the lower hundreds of milliseconds on average. These results demonstrated practical usefulness of our methods in spite of their narrow focus.

In the future, we plan to investigate computational complexity of proposed classes of non-deterministic action reversibility. We also plan to focus on more general subclasses of non-deterministic action reversibility (e.g. strong $S$-reversibility) and we would also like to generalise the results for lifted representation of FOND planning tasks.

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
