# OpenReview forum: "Weak and Strong Reversibility of Non-Deterministic Actions: Universality and Uniformity"
_icaps-conference.org/ICAPS/2024/Conference — ICAPS 2024_

### Official Review · Reviewer_UZ3S · 2024-01-22

**Significance And Importance:** 2
**Soundness:** 3
**Novelty:** 3
**Clarity:** 4
**Overall Evaluation:** 2
**Confidence:** 5

**Weaknesses:**

1: Minor weaknesses that are easily fixable.

**Contributions Of The Paper:**

The paper introduces the concept of action reversibility for non-deterministic actions in the FOND planning setting. In terms of action reversibility, the authors present and focus on two classes of problems: universality and uniformity. To solve these classes of problems, the authors develop a compilation approach that is able to encode the concept of action reversibility into two distinct planning settings: Classical (Deterministic) Planning, to deal with weak reversibility and irreversibility; and FOND Planning, to deal with strong reversibility. The authors also present some formal proofs with respect to the concept of action reversibility in FOND Planning. Experiments and evaluations over well-known FOND Planning benchmarks check the existence of action reversibility in the used benchmarks. Thus, based on the results, the authors identified a type of reversibility for approximately 56.32% of the actions in the used FOND benchmarks.

**Ethical Considerations:**

(1) Not Applicable: The paper does not have any ethical considerations to address

**Nomination For Best Paper:**

No

**Questions For Authors:**

Please, address the following questions in your response/rebuttal.

Q1. How many planning tasks have you used for each of the domains? I couldn't find this information in the paper. Please, clarify that.

Q2.  PRP strong cyclic policies. So, I have two questions regarding this comment.
- Have you modified PRP to extract strong polices?
- If not, have you (simply) ignored the cycles in the extracted polices in your experiments to check strong reversibility?

Q3. My following question may be out of the scope of your paper, but do you see any relation between the presence of action reversibility and irreversibility (and the other notions you proposed in the paper) in the domains' actions and the complexity (and difficulty) of solving problems for these domains? I am asking this because we have seen in recent FOND papers (Geffner and Geffner, 2018, Pereira et al., 2022, and Muise, McIlraith, Beck, 2024) that some FOND planners struggle to find solutions in some domains, especially the ones proposed by Geffner and Geffner in 2018.

**Reproducibility:**

4: Authors promise to release code and domains (whichever apply).

**Strengths Of The Paper:**

The paper is well-written, organised, and easy to follow. The authors introduce the novel concepts by using clear and intuitive examples.

In my view, the key strengths of the paper are as follows:
- The formal concept of action reversibility for non-deterministic actions in the FOND planning;
- A compilation approach that is able to encode the concept of action reversibility into Classical Planning and FOND Planning;
- An empirical evaluation that checks the existence of action reversibility over well-known FOND Planning benchmarks.

**Weaknesses Of The Paper:**

Personally, I see no major weaknesses in the paper, but I do have some minor comments and points for consideration for improving the paper, as follows.

# Minor Comments and Points of Consideration:

C1. The preliminary section (Preliminaries) is really good, but you don't define strong cyclic policies for FOND Planning. Please, include the definition of strong cyclic polices as part of the preliminary section, since it is part of the types of FOND Planning solutions you consider for action reversibility.

C2. In Line 190: What do you mean by the "reverting process"? Do you mean action reversibility? Please, clarify that.

C3. In Line 220: where you define that an action is weakly $S$-reversible, what do you mean by "assuming the "correct" action effect always occurs"? Is that related to (one of) the effects (outcomes) of non-deterministic actions that are able to undo an action? If so, please, clarify that.

C4. What is the difference between Definitions 1 and 2? It seems to me that they broadly define the same thing. Please, clarify that.

C5. In your examples, it would be good if you use a different notation for facts (e.g., $fact$), actions (e.g., $\texttt{action}$), and policies (e.g., $\{fact1, fact2\}: \texttt{action}$). That would improve the readability of the examples.

C6. In Example 2, the action shouldn't be $close-door$ instead of "door-close"?

C7. In Theorem 8, you define $D^0$ as a set of reachable states from the state $s'$. It would be more intuitive to use $S^{s'}$ instead of $D^0$ because $D$ refers to a planning domain.

C9. With respect to your experiments, I have the following (minor) concerns:
- Table 1 could be placed where you comment on the results presented in this table.
- The way you describe the used domains in Table 1 is very confusing (and it is not consistent with the acronyms you describe in Lines 725-730). Moreover, sometimes you use a similar symbol for domain names and action properties (e.g., I. and I).
- How many planning tasks have you used for each of the domains? I couldn't find this information in the paper (see question Q1).

# Writing Issues and Typos:

W1. "i.e." -> "i.e.,".
W2. "... the concept of reverse plans that corresponds to the notion of uniform reversibility..." -> "... the concept of reverse plans, that corresponds to the notion of uniform reversibility...".
W3. There are two sections with the same title, e.g., "Theoretical Properties". Please, consider changing these titles in order to be more informative with respect to the theoretical properties you are presenting and proving.
W4. The sentence in Line 355 doesn't read well.
W5. In Lines 280-285: "applied in any state from the set $S$" -> "applied in any state of the set $S$".
W6. In Example 5 (Lines 360-365): "the policy of union of mentioned policies has no terminal state" -> "the policy of union of the mentioned policies has no terminal state".
W7. "summarised" -> "summarized", "conceptualised" -> "conceptualised", and other occurrences. It seems there's a blend of American and British English in your writing. Pick one and be consistent throughout the paper.
W8. "the resolution of investigated classes" -> "the resolution of the investigated classes".
W9. "strongly uniformly universally" -> Too many adverbs in one sentence.
W10. Line 725: "viz."?!

---

> ### Author Rebuttal · Authors · 2024-01-27
>
> Thank you for all your insightful comments, we'll take them into account!
>
> Q1: For each domain, we have selected only one task. The main purpose of the experiments was to verify whether FOND benchmark domains contain some reversible (rev.) actions and whether the proposed methods are feasible to compute. We further believe that the distribution of grounded actions will not significantly differ among different instances as they share the same lifted domain and actions. Also, the selection of only one instance allows us to be consistent, as for Bus Fare, Climber and River there is only one instance available. We will include that information in the paper.
>
> Q2: Our definition of strong rev. (SR) coincides with strong cyclic solution of Def. 2.10 of Cimatti et al. (2003). We did not distinguish between strong cyclic and acyclic. This fact is explicitly noted on line 200, however, we agree that "strong rev." might suggest "strong (acyclic) solution" instead of "strong cyclic solution". We will clarify that. In our experiments, we used PRP (as is) to generate strong cyclic solutions (albeit either a strong acyclic solution would suffice as a strong reverse policy).
>
> Q3: We would expect that planners that plan through determinised space and that later "close" open states would work better in domains with many SR actions. We believe that with a decreasing number of SR actions, the difficulty of the problem might rise for such FOND planners. Our intuition is supported by the fact that SR actions do not introduce dead-ends (unless they are applied in a state that is already a dead-end). This is further supported by data collected by us and by the mentioned papers, as PRP works well in these domains. However, there are also domains, e.g. Miner, that contain significant amounts of rev. actions, but PRP still struggles to find solutions. Algorithms of this type might also be able to profit from the information that actions are irrev. (as also discussed in the response to Reviewer SSUH, Q1.). FONDSAT seems to struggle on domains with a majority of rev. actions, even though there are also exceptions. This may be caused by the fact that the planner struggles to find an action leading towards a goal state earlier (in comparison to methods working with determinisations). Since this is out of the scope of our paper, this paragraph is based on a brief comparison of the data and claims of the other papers. Yet, the question is interesting and worth a deeper investigation.

---

### Official Review · Reviewer_SSUH · 2024-01-23

**Significance And Importance:** 2
**Soundness:** 3
**Novelty:** 3
**Clarity:** 3
**Overall Evaluation:** 1
**Confidence:** 3

**Weaknesses:**

1: Minor weaknesses that are easily fixable.

**Contributions Of The Paper:**

The paper focuses on action reversibility in the context of non-deterministic planning. It builds on the notion of weak and strong plans from FOND to define corresponding notions of weak and strong reversibility of non-deterministic actions. Besides them, the notions of uniform reversibility and of universal reversibility, as well as their combination, are introduced. A compilation of weak universal uniform reversibility and strong universal uniform reversibility into classical planning and FOND planning, respectively, is provided as well.

**Ethical Considerations:**

(1) Not Applicable: The paper does not have any ethical considerations to address

**Nomination For Best Paper:**

No

**Questions For Authors:**

Taking into account also the outcomes of the experiments, I ask the authors (and myself) whether the proposed notions of reversibility are the most useful/appropriate ones. As a matter of fact, in the conclusion the authors mention the intention of investigating more general classes off non-deterministic action reversibility.

As for the adopted notation/formalism, I wonder whether there are any more compact and natural representations to be used, e.g., automata-based ones.

**Reproducibility:**

3: Authors describe the implementation and domains in sufficient detail.

**Strengths Of The Paper:**

In general, the presented work can be viewed as a generalisation of the results obtained for reversibility in deterministic settings. Even though it cannot be considered a breakthrough in the fields (it needs to be improved in various directions), it seems to me a solid piece of work.

**Weaknesses Of The Paper:**

As for the motivations of the work, I am not sure that the addressed topic is “an important research question”, as the authors write in the introduction (it seems to me that most of the contributions in the literature come from a restricted group of people); surely, it is  topic of interest.

The notation introduced in Section 2 (Preliminaries) is very heavy and not easy to deal with: the single notions are fairly clear, but there are a number of them, and it is not easy to master them all.

From a technical point of view, I did not detect any major flaw. A significant limitation is that complexity issues are not addressed at all (they are left to future work).

The running example looks quite elementary/artificial, and thus not fully convincing.

The quality of the writing is globally satisfactory. There is some punctuation to fix here and there, e.g,, to replace “i.e.” by “i.e.,”

Minor/specific points

Line 145: has {\cal t}^d been explicitly introduced?

Lines 156-158: please check whether everything is ok.

Page 3, Defs. 1 and 2: I would point out that Def. 1 differs from Def. 2 for the exchange of quantifiers: “for each state there exists a policy ..” is replace by: “ there exists a policy such that for each state ..”

Line 332: please check and possibly rewrite the sentence: “let us consider a domain of well-known Transport domain”.

Lines 356-357: please check and possibly rewrite the sentence: “in which the action the policy is reverting”.

Line 359: please check and possibly rewrite the sentence:  “with the only variable with domain”.

Line 845, Giacomo must be replaced by “De Giacomo”.

---

> ### Author Rebuttal · Authors · 2024-01-27
>
> First of all, thank you for your valuable comments. We will take them into account when revising the paper.
>
> 1) As we argue on line 295, uniformity is particularly useful, as if the action is uniformly S-reversible, then we will get by only one policy instead of (at most) |S| different policies. The universality is a desired property as it states that the action is reversible no matter in which state we apply the action. Given that we have information about weak and strong universal uniform reversibility and universal irreversibility of actions of the domain we want to solve, we might prefer some search technique over another. For instance, as long as we apply strongly universally uniformly actions only, we cannot end in a dead-end state (unless we have already been in a dead-end state before). This allows us to prove that techniques such as FF-replan (Yoon, Fern, and Givan 2007) are safe to use up to some point, where more sophisticated techniques might be used to prevent reaching (possible) dead-end states. In further research, we plan to investigate the properties of general S-reversibility, so that S-reversibility may be decided for more actions (especially for columns labeled as "?"). Note that even though the investigated class of reversibility may be perceived as restrictive and specific, the experiments have revealed that a significant number of actions are indeed reversible using those notions.
>
> 2) For the notation/formalism, we mainly took inspiration from Cimatti et al. (albeit in contrast to them, we used SAS+ to represent the environment). The decision was in part motivated by the fact that existing benchmarks are specified in the FOND extension of PDDL and we can leverage the "translate" part of the PRP planner (built on top of FastDownward) to get the representation we consider in the paper. We have not yet explored other representations. Automata-based representation (e.g. De Giacomo & Rubin, IJCAI 2018) might help us in determining more general classes of reversibility. We will explore this possibility, thank you for the suggestion!

---

### Official Review · Reviewer_fKSe · 2024-01-23

**Significance And Importance:** 2
**Soundness:** 3
**Novelty:** 2
**Clarity:** 3
**Confidence:** 4

**Weaknesses:**

0: Minor weaknesses requiring some work to be addressed for the paper to be accepted.

**Contributions Of The Paper:**

This paper approaches reversibility of actions for FOND planning domains. According to the authors, it is the first study on  reversibility for FOND  planning since previous works have focused only on reversibility of classical planning domains. While in the deterministic case an action is reversible if its effects can be undo by a sequence of (other) domain actions, in non-deterministic domains the authors introduce notions of weak and strong reversibility. Briefly, the notion of weakly reversible action refers to “all effects might be undone” and  the notion of strongly reversible action refers to “all the effects can always be undone”. The authors also propose methods to determine whether a non-deterministic action is (weakly or strongly) reversible or irreversible. To evaluate the proposed methods, the authors conducted empirical experiments in benchmark FOND planning domains. In the experiment, the LAMA planner was used to check if an action is irreversible or weak universal uniform reversibility and the PRP planner was used to check strong universal uniform reversibility.

**Ethical Considerations:**

(1) Not Applicable: The paper does not have any ethical considerations to address

**Nomination For Best Paper:**

No

**Overall Evaluation:**

-1: (weak reject)

**Questions For Authors:**

1. A FOND planner can either return a strong, strong-cyclic or "weak policy". Thus, if the FOND planner returns a weak policy can we say that the action is weakly reversible? This would be a theorem not cover by Theorems 11 or 12?

2. For the analyzed domains, it was observed that the majority of actions classified as "weakly universally uniformly reversible" are also classified as "strongly universally uniformly reversible". Is this a particular characteristic of the FOND benchmark domains analysed? If so, could be this characteristic responsable for making possible for the FOND planners to find strong cyclic solutions? It could be an interesting result of this study if you recognize that the most difficult problems for the state-of-the-art FOND planners are the ones with no (or few) strong reversible actions. What do you think about this?

3. Line 628: "Since for weak reversibility, we need only to consider the “correct” effects of actions, we replace each non-deterministic action with its respective determinisations." What you mean by correct? How the algorithm recognize the correct effect of an action?

4. Theorem 12, Line 670:  "... If for each determinisation  there exists a strong goal policy ..." It could be also a strong-cyclic policy?

**Reproducibility:**

4: Authors promise to release code and domains (whichever apply).

**Strengths Of The Paper:**

- The main strengths of this paper is to be the first work on reversibility of non-deterministic actions.
- The paper also defines methods to revert actions in a plan (policy) by applying classical or FOND planners.

**Weaknesses Of The Paper:**

- The algorithms are not properly specified. The authors should provide pseudo-codes instead of defining solutions by Theorem 11 (for weak reversibility detection) and Theorem 12 (for strong reversibility detection).

- Experiments are run over specific instances of each domain without any explanation of why those instances were selected for each domain.

- The concepts "projection of a set Sigma of variable assignments" or "projection of an action" only appears in the paragraph before the experiments and are not properly explained. Thus Theorem 13 is not easy to understand since  it envolves determinization and projection.

- Some domains listed in lines 729-735 are listed in Table 1 with different names. E.g.: Tire World Truck (TI. TRU.) and Tire World (TRI. TI.).

- Typo: In Theorem 11 instead of “e \in eff(e)” should be  “e \in eff(a)”
- Typo: Line 750  “respective Theorems 12 and 13, respectively.”

---

> ### Author Rebuttal · Authors · 2024-01-27
>
> First of all, thank you for your valuable comments. We will take them into account when revising the paper.
>
> Before we answer your questions we want to note that providing the compilations of action reversibility into classical and FOND planning was one of the main targets and contributions of the paper. We believe that the process of the compilation is described in the corresponding theorems and the eventual extraction of the policies is described in the proof sketches, which we believe makes the process straightforward (and hence we have not included pseudocodes).
>
> Now for your numbered questions:
>
> 1) You are partially right: If for each planning task (one for each determinisation) of Theorem 12 we find a weak goal policy, then the action is weakly universally uniformly reversible. However, this situation is a direct consequence of Theorem 11 because such weak goal policies must contain corresponding goal plans (as Theorem 11 specifies for the determinized task).
>
> 2) From the investigated domains and the resulting data it seems that it is a characteristic of the domains. We believe that the fact that action is strongly reversible might be a useful information while generating strong cyclic solutions (as strong reverse policy can be used to "close" undesirable non-deterministic alternatives). Regarding your last question, see the response to reviewer UZ3S, Q3. Thank you for the question!
>
> 3) To deal with non-determinism, we replace a non-deterministic action with deterministic actions, where each non-deterministic effect (of the non-deterministic action) has a corresponding deterministic action (similar to the PRP planner). By "correct" effect we mean those whose corresponding actions are in a goal plan of the determinised task. Such a sequence of determinisations can be transformed into a weak policy for the former non-deterministic task (as shown in the proof sketch of Theorem 11). When we apply the weak policy in the former non-deterministic task, we hope for the "correct" effect, which is the one that corresponds to the determinization of non-deterministic action present in the goal plan of the determinized task. We will try to clarify the sentence and drop the potentially misleading "correct".
>
> 4) As our definition of strong goal policy corresponds to the definition of strong cyclic solutions of Cimatti et al. (2003), there is no difference between strong goal policy and strong cyclic solutions. Please see Q2. of reviewer UZ3S and our response.

---

### Meta-Review · Area_Chair_uMSA · 2024-02-06

**Recommendation:** Accept (Poster)
**Confidence:** 4

**Metareview:**

This paper considers action reversibility of non-deterministic actions, distinguishing weak and strong reversibility, and propose methods how these notions can be tested by means of compilation to classical or FOND planning.
Although the concept of action reversibility is well-understood for classical planning, it is a novel and interesting contribution to study it for FOND planning. This paper is a solid and original first step but its long-term significance is hard to predict (there is no motivating application). The clarity of the paper is overall good but should be improved in some details. The reviewers where satisfied with the rebuttal.

Pros:
- high novelty
- reversibility present in benchmark domains

Cons:
- unclear usefulness of the new notions
- computational complexity left for future work

**Ethical Considerations:**

(1) Not Applicable: The paper does not have any ethical considerations to address